# A Nomogram for Predicting the Possibility of Peripheral Neuropathy in Patients with Type 2 Diabetes Mellitus

**DOI:** 10.3390/brainsci12101328

**Published:** 2022-09-30

**Authors:** Wanli Zhang, Lingli Chen

**Affiliations:** Department of Neurology, The First Affiliated Hospital of Wenzhou Medical University, Wenzhou 325000, China

**Keywords:** diabetic peripheral neuropathy, nomogram, risk factor, type 2 diabetes

## Abstract

Background and Purpose: Diabetic peripheral neuropathy (DPN) leads to ulceration, noninvasive amputation, and long-term disability. This study aimed to develop and validate a nomogram for forecasting the probability of DPN in type 2 diabetes mellitus patients. Methods: From February 2017 to May 2021, 778 patients with type 2 diabetes mellitus were included in this study. We confirmed the diagnosis of DPN according to the Toronto Expert Consensus. Patients were randomly divided into a training cohort (*n* = 519) and a validation cohort (*n* = 259). In the training cohort, univariate and multivariate logistic regression analyses were performed, and a simple nomogram was built using the stepwise method. The receiver operating characteristic (ROC), calibration curve, and decision curve analysis were computed in order to validate the discrimination and clinical value of the nomogram model. Results: About 65.7% and 72.2% of patients were diagnosed with DPN in the training and validation cohorts. We developed a novel nomogram to predict the probability of DPN based on the parameters of age, gender, duration of diabetes, body mass index, uric acid, hemoglobin A1c, and free triiodothyronine. The areas under the curves (AUCs) of the nomogram model were 0.763 in the training cohort and 0.755 in the validation cohort. The calibration plots revealed well-fitted accuracy between the predicted and actual probability in the training and validation cohorts. Decision curve analysis confirmed the clinical value of the nomogram. In subgroup analysis, the predictive ability of the nomogram model was strong. Conclusions: The nomogram of age, gender, duration of diabetes, body mass index, uric acid, hemoglobin A1c, and free triiodothyronine may assist clinicians with the early identification of DPN in patients with type 2 diabetes mellitus.

## 1. Introduction

Diabetic peripheral neuropathy (DPN), a common microvascular complication of type 2 diabetes [1,2], increases the risk of diabetic foot ulceration and non-traumatic lower extremity amputation [3,4]. The primary symptoms of DPN are paresthesia and neuropathic pain in the lower limbs [5,6]. However, almost half of patients present asymptomatically in the early phases of DPN [7], which may result in delayed diagnosis and a high rehospitalization rate. Therefore, timely diagnosis and treatment are essential for preventing the progression and complications of DPN.

An effective diabetic retinopathy prediction model can provide cost-effective and readily accessible risk predictions for patients with type 2 diabetes mellitus [8]. A recent study established different models for predicting DPN in a community healthcare center, and their diagnosis of DPN was not based on nerve conduction studies (NCSs) [9]. Standard diagnostic procedures for DPN incorporate clinical symptoms, physical examination, and NCSs [6,10]. NCSs have been the gold standard for diagnosing DPN, particularly in asymptomatic individuals [11]. Nevertheless, NCSs are expensive and the examination process is unpleasant. In addition, NCSs are not generally available in the majority of Chinese hospitals and are only performed in extreme situations. Therefore, it is vital to design a method that is both straightforward and novel for the early detection of DPN. Thus, the goal of this study was to develop and verify a nomogram for the prediction of DPN in type 2 diabetes patients using cost-effective and readily available characteristics.

## 2. Materials and Methods

### 2.1. Study Population

This cross-sectional study was conducted in the inpatient department of the First Affiliated Hospital of Wenzhou Medical University from February 2017 to May 2021. Participants diagnosed with type 2 diabetes mellitus were continuously enrolled in the study. The diagnosis of type 2 diabetes mellitus was based on the 2017 criteria of the American Diabetes Association (ADA) [12]. All patients underwent neurological assessment and NCSs. We confirmed the diagnosis of DPN by the presence of clinical signs or symptoms associated with neuropathy and abnormal electromyography tests, according to the Toronto Expert Consensus [10]. The participants were divided into DPN and non-DPN groups. The inclusion criteria were: (1) age ≥18 years, (2) patients with type 2 diabetes mellitus. Exclusion criteria included other causes of peripheral neuropathy, malignant tumor, acute infectious disease, severe liver or renal disease, heart failure, metabolic disease (thyroid disease or vitamin B12 deficiency), other severe life-shortening illness, and any medication that could affect serum uric acid. The study was approved by the ethics committee of the First Affiliated Hospital of Wenzhou Medical University (NO.KY2021-R141). Written informed consent was obtained from patients and their relatives.

### 2.2. Peripheral Neuropathy Assessment

All subjects accepted neurological assessment using the neuropathy symptom score (NSS) and neuropathy disability score (NDS) [13,14,15]. The NSS included burning pain, tingling, numbness, fatigue, cramping, or aching in the legs. The NDS included temperature perception, vibration, pinprick sensation, and Achilles reflex. The clinical criteria for peripheral neuropathy are NDS scores of ≥6 or NDS scores of 3–5, with NSS scores of ≥5 [16].

Electrophysiological examinations were carried out by experienced technicians with the electromyography instrument (Kipoint-4 type, Vidi; NDI-200P + type; Poseidon). Patients kept limb skin temperature between 32 to 35 ℃ during testing. The compound muscle action potential (CMAP) amplitude, distal latency, and conduction velocity (CV) of bilateral ulnar, median, tibial, and common peroneal nerves were measured for motor nerves. The sensory nerve action potential (SNAP) amplitude and CV of bilateral ulnar, median and superficial peroneal nerves were measured for the sensory nerve. Bilateral measurements of the F-wave latency of the tibial nerve were also calculated. The data of healthy individuals in the neurophysiology laboratory of Peking Union Medical College Hospital were used as reference values. Electrophysiological experts judged abnormal NCSs based on the abnormality of one or more attributes in two or more nerves [17]. In addition, the NCSs parameters of the more severe side were collected. The mean motor nerve amplitude (MNAmp) was calculated as (ulnar nerve motor amplitude + median nerve motor amplitude + tibial nerve motor amplitude + common peroneal nerve motor amplitude)/4. A similar formula was employed to calculate the mean motor nerve conduction velocity (MNCV), mean sensory nerve amplitude (SNAmp), and mean sensory nerve conduction velocity (SNCV) [18].

### 2.3. Clinical and Laboratory Data Collection

Information about medical history, physical examination, and clinical data, including age, gender, duration of diabetes (duration), body mass index (BMI), history of smoking, hypertension, and dyslipidemia, were collected from electronic medical records. 

After 8–10 hours of overnight fasting, blood samples were obtained in the morning. Routine blood tests, including neutrophils, lymphocytes, and platelet (PLT), were determined by an autoanalyzer (Mindray BC6800, Shenzhen, China). NLR was defined as the ratio of neutrophil count to lymphocyte count. Hemoglobin A1c (HbA1c) was measured by high-performance liquid chromatography. Uric acid (UA), fasting plasma glucose (FPG), total cholesterol (TC), triglyceride, high-density lipoprotein cholesterol (HDL-C), and low-density lipoprotein cholesterol (LDL-C) were determined by an automated chemistry analyzer (Beckman AU5800, Indianapolis, IN, USA). Free triiodothyronine (FT3), free thyronine (FT4), and thyroid stimulating hormone (TSH) were measured by an automatic chemiluminescence analyzer (Beckman DXI800, Indianapolis, IN, USA). Fibrinogen (FIB) was determined by an automated coagulation analyzer (Stago STA-R Max, Paris, France).

### 2.4. Training and Validation of the Nomogram

Finally, 778 patients with type 2 diabetes were included in this study. The patients were randomly divided into two groups (two patients in the training cohort and one in the validation cohort).

### 2.5. Statistical Analysis

We performed the Shapiro–Wilk test to determine whether the variable conformed to the normal distribution. Continuous variables with normal distributions were represented as mean ± standard deviation (SD), and continuous variables with skewed distribution were described as median ± interquartile ranges. Student’s *t*-tests or the Kruskal–Wallis test were used to compare the difference between two groups in continuous variables. Categorical variables were represented as frequencies (percentages) and compared using Chi-square tests between the two groups. 

The comparison of baseline characteristics of the training cohort stratified by the presence of DPN were presented. First, all variables were assessed using univariate analysis between the DPN group and the non-DPN group, and univariate logistic regression was used to identify the risk factors of DPN in the training cohort. Secondly, the stepwise multivariate logistic regression analyses were performed with *p* < 0.05 variables from the univariate logistic regression. According to previous studies, the clinical variables recognized as critical factors were also entered into the multivariate logistic regression analyses. Variables with *p* < 0.05 in the multivariable analysis were retained to establish a nomogram in the training group. The score for each variable was calculated based on the regression coefficient values.

The validation of the nomogram model consisted of two parts. Initially, the areas under the curve (AUC) of the receiver operating characteristic (ROC) were used to evaluate the nomogram's discrimination capacity in the training cohort. We also conducted the AUC with a 95% CI using 500 bootstrap resamplings for internal validation. The calibration curve was assessed graphically by smoothing a scatter plot of the predicted and actual probabilities. Decision curve analysis (DCA) was performed in order to determine the clinical net benefit of the model. Second, the ROC, calibration curve, and DCA were also performed in the validation cohort. Correlations between variables in the nomogram and NCSs parameters were assessed using Pearson's or Spearman's test in all of the patients. Then, multiple linear regression analysis was executed in order to identify the relationship between variables in the nomogram and NCS parameters. Two-sided *p* < 0.05 was considered statistically significant. All of the statistical analyses were performed with statistical packages R version 3.5.1(https://www.R-project.org, access date on 20 August 2022) and SPSS Version 24.0 (IBM, Chicago, IL, USA).

## 3. Results

Seven hundred and seventy-eight eligible patients with type 2 diabetes were randomly divided into a training cohort (*n* = 519) and a validation cohort (*n* = 259). About 65.7% and 72.2% of patients were diagnosed with DPN in the training and validation cohorts. The differences in baseline characteristics between these two were insignificant (Table 1).

### 3.1. Univariate and Multivariate Analyses

In order to identify the risk factors of DPN, a univariate analysis was performed between the DPN group and non-DPN group in the training cohort (Table 2). Significant differences in the following variables were obtained: gender, age, duration, smoking, hypertension, UA, FIB, HbA1c, and FT3. 

According to previous studies, BMI was also recognized as an essential risk factor for DPN. Therefore, we included BMI in the multivariable logistic regression analysis. As shown in Table 3, gender, age, duration, BMI, UA, HbA1c, and FT3 were independently associated with the presence of DPN.

### 3.2. Nomogram Development and Validation

Based on the multivariable logistic regression analysis, the nomogram was developed for predicting DPN risk based on gender, age, duration, BMI, UA, HbA1c, and FT3 in the training cohort (Figure 1). Data were collected in patients with type 2 diabetes, and the position of each variable on the corresponding axis was confirmed. A vertical line was drawn from each variable’s position to the top “points” axis to collect the variable’s score. Then, the users added up the score of each variable to acquire the total score on “total points”. They then drew a vertical line from the total points axis to the bottom scale to assess the DPN risk. “For example, a 70-year-old (60 points) male (20 points) patient sufferers from a 5-year history of type 2 diabetes (10 points), has 30 kg/m^2^ of BMI (10 points), 500 umol/L of UA (30 points), 7 of HbA1c (10 points), and 8 pmol/L of FT3 (50 points). He receives a total score of 190 points by adding all the points. The estimated probability of DPN for this patient is less than 50%.

The AUC of the nomogram was 0.763 for the training cohort (Figure 2), and the internal validation by 500 bootstrap resamplings was 0.759. The AUC of the nomogram was 0.755 for the validation cohorts (Figure 2), presenting good predictions. Furthermore, the predictive performance of the nomogram model for different subgroups in the validation cohorts was also quantified (Table 4), indicating that the nomogram model was an effective classifier in different subgroups. The nomogram calibration plots revealed moderate prediction accuracy in the training and validation cohorts (Figure 3). Figure 4 shows the DCA for the training and validation cohorts to predict the possibility of DPN. A line with greater distance between the model curve and the black and gray denotes a better clinical value for the nomogram. 

### 3.3. Relationship between Variables and NCSs Parameters

In Appendix A, except for F-wave, the improvement of other NCSs parameters represented better nerve conduction function. In correlation and multivariate linear regression analyses, age, duration, UA, and HbA1c were inversely correlated with NCSs parameters (except F-wave). Otherwise, BMI and FT3 were positively correlated with NCSs parameters (except F-wave).

## 4. Discussion

In the present study, we established a practical and simple nomogram to predict the probability of DPN tailored to individual patients. The predictive nomogram showed good discriminatory strength and moderate clinical value in training and validation cohorts. Our nomogram is based on gender, age, duration, BMI, UA, HbA1c, and FT3 to provide a user-friendly and convenient tool for clinical practice. In addition, the performance of the nomogram was beneficial and stable in various subgroups. Furthermore, the independent factors in the nomogram were also independently associated with NCS parameters.

The diagnostic criteria of DPN varied in different studies [9,19,20,21]. Our study confirmed the diagnosis of DPN with nerve conduction. NCS is proposed as a gold standard for early subclinical DPN diagnosis [10]. However, NCS is uncomfortable, expensive, and not widely used in most Chinese hospitals. We developed a user-friendly nomogram to predict DPN, which alleviated both the mental burden and financial costs. Another study focused solely on the community population and established a different nomogram based on the perspective of statistical research [9]. In addition, their diagnostic criteria of DPN were based on the Toronto clinical scoring system score, which might bring about false positives [9]. The risk factors in our study were different from previous studies, especially since few studies have identified the correlation between FT3 and DPN [9]. To the best of our knowledge, our nomogram is the first prediction model for DPN based on routine biochemical indexes and lifestyle factors for inpatients in the Chinese tertiary hospital.

The risk factors included in this nomogram are similar to the results of other prediction models. Advancing age is a nonmodifiable independent predictor for DPN [16]. Many studies indicated that peripheral neurodegeneration was related to aging [22,23]. Our analysis also demonstrated that increasing age was positively associated with DPN. Male patients were more likely to suffer from type 2 diabetes over the course of a 3-year follow-up [24]. In this current report, male patients were observed to have a higher possibility of developing DPN. A higher cumulative incidence of DPN was observed in male than female patients in young adults [25]. However, another study with Iranian participants came to an opposite conclusion [26]. Numerous studies, including our research, indicated that T2DM patients with longer disease duration were more likely to develop DPN [9,27,28].

The present study also investigated the associations between DPN and other important risk factors. The relationship between BMI and DPN is still controversial. Previous studies have suggested that BMI was positively linked to DPN [9,25]. Nevertheless, one study indicated that lower BMI was a high independent risk factor for DPN [29]. Another study demonstrated that the relationship between BMI and DPN was U-shaped [30]. Our findings confirmed that a low BMI was associated with a higher DPN risk. Hyperuricemia was reported to be related to endothelial dysfunction and oxidative stress [31,32]. Our study showed that hyperuricemia was connected to an increased DPN risk, consistent with other studies [26,33]. Nevertheless, another cross-sectional study described a contrary relationship between large-nerve fiber dysfunction and UA levels in diabetic patients [34].

Poor glycemic control, such as increased HbA1c, was related to a higher DPN risk in the TODAY cohort [25]. We observed that a 1-unit increase in HbA1c led to an increase of 27% in the possibility of developing DPN. The active form of T3 bonded to the thyroid receptors to initiate the expression of the myelination-associated genes [35]. Abnormal serum thyroid hormone levels have been observed for decades in patients with T2DM [36]. Only one study has identified a positive correlation between FT3 and NCSs [37]. The present study demonstrated that a higher FT3 level was related to a lower DPN risk.

The present study has some remarkable strengths. Our study demonstrated good discrimination and predictability of the nomogram for DPN with factors that can be readily obtained in general healthcare settings. The predictive power of the nomogram model was suitable and stable in various clinical subgroups, suggesting that it could be widely used in clinical practice. NCSs were considered a gold standard for the diagnosis of DPN. Moreover, the risk factors of the nomogram were independently associated with NCS parameters, indicating a close connection between the nomogram model and nerve conduction function.

First, this was a single-center study that was only applicable to patients with type 2 diabetes and could not be applied to communities of type 1 diabetes patients. Second, we did not collect information on other complications of T2DM and genetic markers. These tests may, however, increase the cost of DPN screening and other clinical indicators. 

## 5. Conclusions

In conclusion, the nomogram model is composed of age, gender, duration of diabetes, BMI, UA, HbA1c, and FT3, has a good predictive ability of DPN in patients with type 2 diabetes mellitus. Since NCSs are uncomfortable and expensive, developing a tool for limited healthcare resource providers in rural areas is beneficial. Due to its simplicity and applicability, this nomogram might be generalized for clinical practice and identifies patients with high DPN risk at the early stage.

## Figures and Tables

**Figure 1 brainsci-12-01328-f001:**
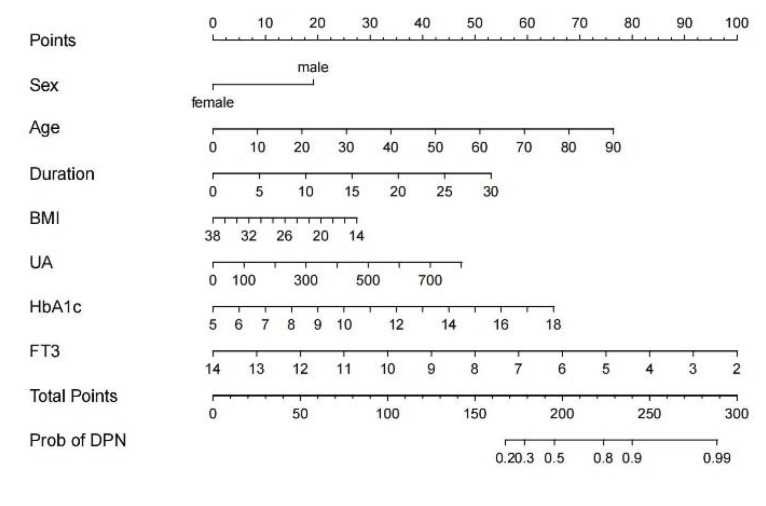
The nomogram prediction model for DPN risk. Data were collected in patients with type 2 diabetes, and the position of each variable on the corresponding axis was confirmed. A vertical line was drawn from each variable’s position to the top “Points” axis to collect the variable’s score. Then, the users added up the score of each variable to acquire the total score on “Total Points”. They then drew a vertical line from the total points axis to the bottom scale to assess the DPN risk. For example, a 70-year-old (60 points) male (20 points) patient sufferers from a 5-year history of type 2 diabetes (10 points), has 30 kg/m^2^ of BMI (10 points), 500 umol/L of UA (30 points), 7 of HbA1c (10 points), and 8 pmol/L of FT3 (50 points). He receives a total score of 190 points when all of the points are added. The estimated probability of DPN for this patient was calculated. DPN, Diabetic peripheral neuropathy; BMI, body mass index; UA, uric acid; HbA1c, hemoglobin A1c; FT3, free triiodothyronine.

**Figure 2 brainsci-12-01328-f002:**
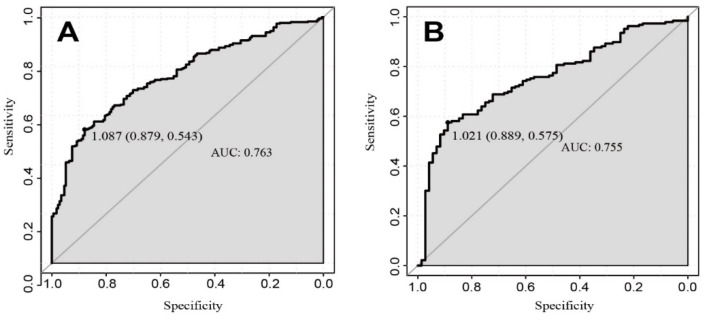
ROC curves of the nomogram in training (**A**) and validation (**B**) cohort. The black dot represents the best cut-off value: 1.087 for the training cohort and 1.021 for the validation cohort.

**Figure 3 brainsci-12-01328-f003:**
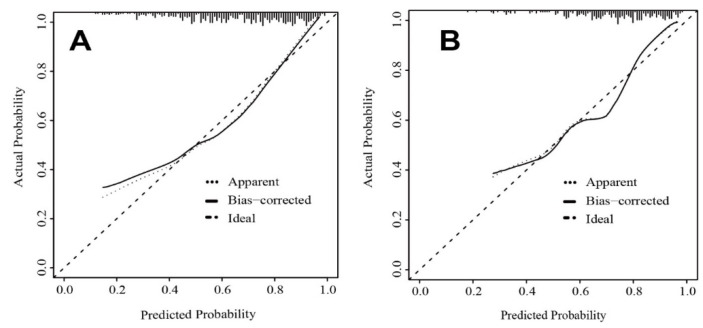
Calibration plot. The x-axis represents the nomogram-predicted probability, and the y-axis represents the actual probability of diabetic peripheral neuropathy. (**A**) A nomogram calibration plot in the training cohort. A perfect prediction would fall along the 45-degree line (“ideal” line). The “apparent” line represents the training cohort, and the solid black line represents bias corrected by bootstrapping (500 repetitions), indicating observed nomogram performance. (**B**) A nomogram calibration plot in the validation cohort. A perfect prediction would fall along the 45-degree line (“ideal” line). The “apparent” line represents the validation cohort, and the solid black line represents bias corrected by bootstrapping (500 repetitions), indicating observed nomogram performance.

**Figure 4 brainsci-12-01328-f004:**
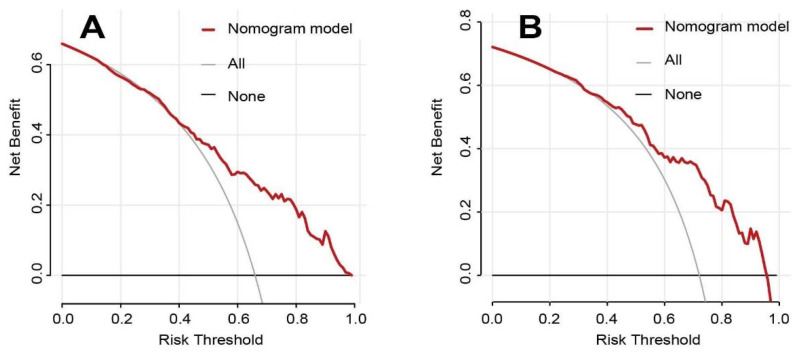
The decision curve analysis of the nomogram predicts the possibility of DPN in training (**A**) and validation (**B**) groups. The horizontal solid black line represents the net benefit when no patient was considered to have DPN. The gray line represents the net benefit when all patients were identified as suffering from DPN. The area along the red line (nomogram model line), the black line, and the gray line representing the net benefit of the nomogram was significantly higher than that of the "no patient" and " all patients" schemes, suggesting that the nomogram has good clinical applicability.

**Table 1 brainsci-12-01328-t001:** The comparison of baseline characteristics of the patients in training and validation cohorts.

Variables	Training Cohort (*n* = 519)	Validation Cohort (*n* = 259)	*p* Value
DPN, no. (%)	341 (65.7%)	187 (72.2%)	0.081
Age (years)	57.76 ± 12.95	58.97 ± 12.49	0.265
Male, no. (%)	334 (64.4%)	159 (61.4%)	0.465
Smoking, no. (%)	180 (34.7%)	77 (29.7%)	0.193
Hypertension, no. (%)	266 (51.3%)	143 (55.4%)	0.307
Dyslipidemia, no. (%)	168 (32.4%)	78 (30.1%)	0.579
Duration (years)	10 (4–12)	10 (5–15)	0.061
BMI (kg/m^2^)	24.21 ± 3.49	24.23 ± 3.60	0.803
FPG (mmol/L)	7.95 ± 2.90	8.10 ± 3.69	0.813
UA (umol/L)	329.04 ± 101.10	326.47 ± 98.71	0.748
TC (mmol/L)	4.79 ± 1.35	4.81 ± 1.48	0.835
TG (mmol/L)	1.91 ± 1.74	1.89 ± 1.47	0.481
HDL-C (mmol/L)	1.02 ± 0.29	1.05 ± 0.43	0.753
LDL-C (mmol/L)	2.61 ± 0.98	2.58 ± 1.03	0.629
FIB (g/L)	3.79 ± 1.24	3.71 ± 1.05	0.950
NLR	2.36 ± 1.41	2.45 ± 1.62	0.438
PLT (×10^9^/L)	221.10 ± 64.93	222.26 ± 61.99	0.809
HbA1c (%)	9.32 ± 2.38	9.42 ± 2.82	0.940
TSH (mIU/L)	1.34 (0.91–2.03)	1.31 (0.86–1.92)	0.124
FT4 (pmol/L)	11.24 ± 2.33	11.34 ± 2.11	0.423
FT3 (pmol/L)	4.64 ± 0.79	4.67 ± 0.84	0.568

DPN, Diabetic peripheral neuropathy; BMI, body mass index; FPG, fasting plasma glucose; UA, uric acid; TC, total cholesterol; TG, triglyceride; HDL-C, high-density lipoprotein cholesterol; LDL-C, low-density lipoprotein cholesterol; FIB, fibrinogen; NLR, the ratio of neutrophil count to lymphocyte count; PLT, platelet; HbA1c, hemoglobin A1c; TSH, thyroid stimulating hormone; FT4, free thyronine; FT3, free triiodothyronine.

**Table 2 brainsci-12-01328-t002:** The comparison of baseline characteristics according to the presence of DPN and the univariate logistic regression analysis for DPN in the training cohort.

Variables	Training Cohort			Univariate Logistic Regression Analysis	
	DPN Group (*n* =341)	Non-DPN Groups (*n* =178)	*p* Value	OR (95% CI)	*p* Value
Male, no. (%)	235 (68.9%)	99 (55.6%)	0.004	1.769 (1.217–2.572)	0.003
Age (years)	59.57 ± 12.36	54.31 ± 13.40	<0.001	1.032 (1.017–1.047)	<0.001
Duration (years)	10.00 (5.00–15.00)	6.50 (2.00–10.00)	<0.001	1.089 (1.056–1.124)	<0.001
Smoking, no. (%)	130 (38.1%)	50 (28.1%)	0.029	1.577 (1.064–2.337)	0.023
BMI (kg/m^2^)	24.03 ± 3.22	24.56 ± 3.93	0.128	0.958 (0.909–1.009)	0.105
Hypertension, no. (%)	189 (55.4%)	77 (43.3%)	0.011	1.631 (1.132–2.350)	0.009
Dyslipidemia, no. (%)	114 (33.4%)	54 (30.3%)	0.538	1.153 (0.783–1.712)	0.475
FPG (mmol/L)	8.08 ± 3.01	7.68 ± 2.67	0.123	1.050 (0.984–1.121)	0.137
UA (umol/L)	337.95 ± 106.03	311.97 ± 88.71	0.003	1.003 (1.001–1.005)	0.006
TC (mmol/L)	4.74 ± 1.37	4.88 ± 1.30	0.266	0.928 (0.812–1.061)	0.274
TG (mmol/L)	1.87 ± 1.33	1.99 ± 1.43	0.523	0.962 (0.869–1.065)	0.461
HDL-C (mmol/L)	1.03 ± 0.29	1.01 ± 0.29	0.491	1.253 (0.660–2.378)	0.490
LDL-C (mmol/L)	2.58 ± 1.03	2.67 ± 0.88	0.280	0.904 (0.753–1.086)	0.280
FIB (g/L)	3.90 ± 1.32	3.58 ± 1.06	0.003	1.260 (1.068–1.487)	0.006
NLR	2.41 ± 1.41	2.27 ± 1.40	0.259	1.082 (0.942–1.243)	0.262
PLT (×10^9^/L)	220.54 ± 65.16	222.17 ± 64.64	0.786	1.000 (0.997–1.002)	0.786
HbA1c (%)	9.53 ± 2.36	8.91 ± 2.36	0.004	1.122 (1.036–1.215)	0.005
TSH (mIU/L)	1.35 (0.91–2.02)	1.33 (0.94–2.22)	0.941	1.048 (0.927–1.185)	0.457
FT4 (pmol/L)	11.36 ± 2.41	11.03 ± 2.16	0.112	1.069 (0.981–1.165)	0.127
FT3 (pmol/L)	4.55 ± 0.82	4.81 ± 0.69	<0.001	0.642 (0.495–0.834)	0.001

BMI, body mass index; FPG, fasting plasma glucose; UA, uric acid; TC, total cholesterol; TG, triglyceride; HDL-C, high-density lipoprotein cholesterol; LDL-C, low-density lipoprotein cholesterol; FIB, fibrinogen; NLR, the ratio of neutrophil count to lymphocyte count; PLT, platelet; HbA1c, hemoglobin A1c; TSH, thyroid stimulating hormone; FT4, free thyronine; FT3, free triiodothyronine.

**Table 3 brainsci-12-01328-t003:** Multivariate logistic regression analysis for risk factors associated with DPN in the training cohort.

Variables	Multivariable Analysis		*p* Value
	OR	(95% CI)	
Gender	2.607	1.658–4.138	<0.001
Age (year)	1.040	1.022–1.060	<0.001
Duration (year)	1.091	1.053–1.132	<0.001
BMI (kg/m^2^)	0.939	0.884–0.996	0.037
UA (umol/L)	1.003	1.001–1.005	0.012
HbA1c (%)	1.267	1.151–1.402	<0.001
FT3 (pmol/L)	0.674	0.505–0.876	0.005

BMI, body mass index; UA, uric acid; HbA1c, hemoglobin A1c; FT3, free triiodothyronine.

**Table 4 brainsci-12-01328-t004:** ROC analysis of the nomogram model in different subgroups of the validation cohorts.

Subgroups	AUC
Gender	
Male	0.769
Female	0.715
Age (year)	
≥65 (years)	0.762
<65 (years)	0.736
Duration (year)	
>10 years	0.802
≤10 years	0.681
BMI	
≥24 (kg/m^2^)	0.763
<24 (kg/m^2^)	0.723
Hypertension	
Hypertension	0.788
No hypertension	0.694

ROC, receiver-operating characteristic; BMI, body mass index.

## Data Availability

All data that support the findings of this study are available upon request from the corresponding author.

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
