# Peer review of "A Nomogram for Predicting the Possibility of Peripheral Neuropathy in Patients with Type 2 Diabetes Mellitus"

_brainsci, 2022, doi:10.3390/brainsci12101328_

Round 1

Reviewer 1 Report

Dear Authors,

thank you for giving me the opportunity to read this your manuscript entitled "A nomogram for predicting the possibility of pheripheral neuropathy among type 2 diabetes mellitus patients", submitted for publication in Brain Sciences. 

I have just one major comment.   

In 2021, some Chinese investigators published a study on "risk factors of pheripheral neuropathy in type 2 diabetes mellitus and establishment of prediction model". You reported this as reference no. 9 (https://doi.org/10.4093/dmj.2020.0100).  Your manuscript seemed similar to this. 

Please, clarify the subsstancial differences between this study and yours. Your study was performed in the inpatient department of an university hospital, and the other study was carried out in a community. It is obvious that this can not be the mean (or the only) element of difference. As you correctly highlighted in line 241, the casuality in your study was very limited. In addition, you proposed your nomogram in general healthcare settings, just like the other Chinese study. 

What is the originality and usefulness of your study ? I do not understand and I would ask you to remove these concerns. 

Author Response

Responses to the Reviewer 1
Question 1: Dear Authors, thank you for giving me the opportunity to read this your manuscript entitled "A nomogram for predicting the possibility of peripheral neuropathy among type 2 diabetes mellitus patients", submitted for publication in Brain Sciences. I have just one major comment. In 2021, some Chinese investigators published a study on "risk factors of peripheral neuropathy in type 2 diabetes mellitus and establishment of prediction model". You reported this as reference no. 9 (https://doi.org/10.4093/dmj.2020.0100). Your manuscript seemed similar to this. Please, clarify the subsstancial differences between this study and yours. Your study was performed in the inpatient department of an university hospital, and the other study was carried out in a community. It is obvious that this can not be the mean (or the only) element of difference. As you correctly highlighted in line 241, the casuality in your study was very limited. In addition, you proposed your nomogram in general healthcare settings, just like the other Chinese study. What is the originality and usefulness of your study? I do not understand and I would ask you to remove these concerns. 
Answer: Thanks for your valuable suggestions. We appreciated it very much! Nowadays, most studies, including “Study on Risk Factors of Peripheral Neuropathy in Type 2 Diabetes Mellitus and Establishment of Prediction Model,” diagnose DPN only based on the Toronto clinical scoring system score, which might bring about false positives. Our study confirmed the diagnosis of DPN based on nerve conduction studies (NCSs). NCSs was proposed as a gold standard for early diagnosis of DPN. We have reason to believe that our DPN diagnostic method can be more accurately diagnose DPN than other studies. Second, the risk factor in our study differed from previous studies, especially since few studies have identified the correlation between FT3 and DPN. To the best of our knowledge, our nomogram is the first prediction model for DPN based on routinely biochemical indexes and lifestyle factors for inpatients in the Chinese tertiary hospital (We have corrected this in our discussion section). 
The purpose of our study was to create and validate a nomogram for the prediction of DPN with cost-effective and available parameters in type 2 diabetes mellitus patients. NCSs was considered a gold standard for the diagnosis of DPN. The risk factors in the nomogram were also independently associated with NCSs parameters, indicating a close connection between the nomogram model and nerve conduction function in our study. Due to NCSs were uncomfortable and expensive, developing a tool for inexperienced healthcare and limited healthcare resources providers in rural areas is beneficial. If the nomogram indicates the patients have a high probability of DPN, further NCSs examination may be necessary. To some extent, this can reduce the waste of medical resources.
Discussion section (page 9, line 267): Furthermore, the risk factors in our study were different from previous studies, especially since few studies have identified the correlation between FT3 and DPN.

Reviewer 2 Report

Title : “A Nomogram for Predicting the Possibility of Peripheral Neuropathy Among Type 2 Diabetes Mellitus Patients”

In this work the authors provide a nomogram to predict the probability of DPN tailored to individual patients. The authors claim that this predictive nomogram showed good discriminatory strength and moderate clinical value in training and validation cohorts. In particular, this nomogram was based on male, age, duration, BMI, SUA, HbA1c, and FT3 to provide a user-friendly and convenient tool for clinical practice.

General comment: Although the topic of this work is interesting and the use of a nomogram may assist clinicians in a favourable way, this work should be reworked to improve its quality and impact. In particular, the readability of the main text should be enhanced providing, at the beginning of the main text, a table where all the used acronyms are described without a big effort of the readers, who is now forced to go back and forth to find the definition of each acronyms inside the main text. Then, the statistical methods used to build nomograms should be better explained for interested readers without of strong statistical background. In addition, the use of nomograms should be also explained in a better way through examples of predictions. Finally, the value of this work is not totally clear since some important limitations are reported by the authors in the Discussion section.

Some detailed comments:

Lines: “The study was approved by the ethics com- 66
mittee of the First Affiliated Hospital of Wenzhou Medical University (NO.KY2021-R141). 67
Written informed consents were obtained from patients or their representatives “

*) The informed consent should be obtained by all people who are involved in the experimental trials. It is not clear who “their representative” may be. Please explain.

Lines: “2.4. Statistical Analysis 111

Continuous variables described as the means with standard deviations or medians 112

with interquartile ranges were compared between training and validation groups using 113

Students’t-tests or the Kruskal–Wallis test. Categorical variables reported as percentages 114

were compared between the two groups using Chi-square tests. First, univariate analysis 115

and univariate logistic regression were performed to identify the risk factors of DPN in 116

the training group. Then the clinical variables reported as essential factors in previous 117

studies were also selected. Secondly, the stepwise multivariate logistic regression analyses 118

were performed using variables with P < 0.05 from univariate analyses and critical clinical 119

variables. A nomogram was established based on independent factors selected by multi- 120

variate logistic regression analyses of the training cohort. The area under the receiver- 121

operating characteristics curve (AUC) was used to evaluate the nomogram's discrimina- 122

tion capacity. And we also conducted the AUC with a 95% CI using 200 bootstrap 123

resamplings for internal validation. A calibration curve was assessed graphically by 124

smoothing a scatter plot of the predicted and actual probabilities. Decision curve analysis 125

(DCA) was performed to determine the clinical net benefit of the model. The performance 126

of the nomogram was further evaluated in the validation cohort via receiver-operating 127

characteristic (ROC), calibration curve, and DCA. Correlations between independent fac- 128

tors and NCSs parameters were assessed using Pearson's or Spearman's test and multiple 129

linear regression analysis. Two-sided P < 0.05 was considered statistically significant. All 130

the statistical analyses were performed with statistical packages R (http://www.R-pro- 131

ject.org) and SPSS Version 24.0 (SPSS Inc., Chicago, IL, USA)”

*) These lines are not too clear and should be reworked in order to better clarify the statistical procedures used in this work.

In particular,

lines “Continuous variables described as the means with standard deviations or medians with interquartile ranges”

* What are the variables assessed through their mean and standard deviation ? What are the variables assessed through their medians and interquartile ranges. Why they are different ? What is the test used to investigate the nature of these variables (paramateric, non parametric) and of their distributions (normal, not normal).

Lines: “compared between training and validation groups using Students’t-tests or the Kruskal–Wallis test”

*)When the Student and the Kruskal-Wallis tests were used ? What were the main hypotheses to do this ? A clear distinction between parametric and not parametric tests should be provided together with the list of all the variables involved in these test (with the explicit indication of their nature).

Lines: “Categorical variables reported as percentages 114

were compared between the two groups using Chi-square tests. First, univariate analysis 115

and univariate logistic regression were performed to identify the risk factors of DPN in 116

the training group. Then the clinical variables reported as essential factors in previous 117

studies were also selected. Secondly, the stepwise multivariate logistic regression analyses 118

were performed using variables with P < 0.05 from univariate analyses and critical clinical 119

variables.”

*) Please explain in detail the comparison trough the Chi-square test and the main hypotheses.

*) Please explain in detail the “univariate analysis” and the “univariate logistic regression” together with the main hypotheses.

*) Please explain in a better way and provide all the needed details for the lines: “Secondly, the stepwise multivariate logistic regression analyses were performed using variables with P < 0.05 from univariate analyses and critical clinical variables.”

lines: “A nomogram was established based on independent factors selected by multi- 120

variate logistic regression analyses of the training cohort. The area under the receiver- 121

operating characteristics curve (AUC) was used to evaluate the nomogram's discrimina- 122

tion capacity. And we also conducted the AUC with a 95% CI using 200 bootstrap 123

resamplings for internal validation. A calibration curve was assessed graphically by 124

smoothing a scatter plot of the predicted and actual probabilities.”

*) Please explain in all details how the nomogram has been built. The authors should also explain why “we also conducted the AUC with a 95% CI using 200 bootstrap resamplings for internal validation”. Why just 200 bootstrap ?

Lines: “A calibration curve was assessed graphically by 124

smoothing a scatter plot of the predicted and actual probabilities. Decision curve analysis 125

(DCA) was performed to determine the clinical net benefit of the model. The performance 126

of the nomogram was further evaluated in the validation cohort via receiver-operating 127

characteristic (ROC), calibration curve, and DCA.”

*) Please explain better in all details

lines: “Decision curve analysis (DCA) was performed to determine the clinical net benefit of the model. The performance 126

of the nomogram was further evaluated in the validation cohort via receiver-operating 127

characteristic (ROC), calibration curve, and DCA. Correlations between independent fac- 128

tors and NCSs parameters were assessed using Pearson's or Spearman's test and multiple 129

linear regression analysis. Two-sided P < 0.05 was considered statistically significant. All 130

the statistical analyses were performed with statistical packages R (http://www.R-pro- 131

ject.org) and SPSS Version 24.0 (SPSS Inc., Chicago, IL, USA)”

*) Please explain better. All the statistical analyses should be first explained and then performed with statistical packages in R.

lines: “A total of 778 eligible patients with type 2 diabetes were randomly divided into a 134
training cohort (n = 519) and a validation cohort (n = 259). About 65.7% and 72.2% of pa- 135
tients were diagnosed as DPN in the training and validation cohorts. The differences in 136
baseline characteristics between the training and validation cohorts were insignificant 137
(Table 1). “

*) Table 1 is not clear. Please improve together with the caption, which should be more descriptive.

Lines: “3.1. Univariate and Multivariate Analyses 139

To identify the risk factors of DPN, a univariate analysis was performed between the 140

DPN group and non-DPN groups in the training cohorts (Table 2). Significant differences 141

in the following variables were obtained: male, age, duration, smoking, hypertension, 142

SUA, FIB, HbA1c, and FT3. “

*) Table 2 is not clear. Please improve together with the caption, which should be more descriptive.

Table 3. Multivariate logistic regression analysis for DPN risk factors in the training cohort.

*) see the previous comment.

Lines :”3.2. Nomogram Development and Validation 151

Based on the multivariable logistic regression analysis, the nomogram was devel- 152

oped for predicting the possibility of DPN by using male, age, duration, BMI, SUA, 153

HbA1c, and FT3 in the training cohort (Figure 1). Each variable's position on the corre- 154

sponding “Points” axis was identified to predict the DPN's probability. Then the users 155

added up the score of each variable to acquire the total score. By drawing a vertical line 156

from the total points axis to the bottom scale to assess the probability of DPN. 15”

and

lines: “Figure 1. The nomogram prediction model for DPN risk. The position of each variable on the corre- 159

sponding “Points” axis was identified. Then the users added up the score of each variable to acquire 160

the total score. By drawing a vertical line from the total points axis to the bottom scale to assess the 161

probability of DPN. DPN, Diabetic peripheral neuropathy; BMI, body mass index; SUA, serum uric 162

acid; HbA1c, glycated hemoglobin; FT3, free triiodothyronine.”

*) This important part of the main text should be reworked to improve its clarity. In particular, it is not clear how the single scales correlated to the total score. The authors should provide an example to allow the interested readers to better understand the procedure. The authors may use arrows as standard in nomograms.

Lines: “The AUC of the nomogram was 0.763 for the training cohort (Figure 2), and the in- 164

ternal validation by 200 bootstrap resamplings was 0.760. The AUC of the nomogram was

0.755 for the validation cohorts (Figure 2), presenting with good prediction. Furthermore, 166

the predictive performance of the nomogram model for different subgroups in the valida- 167

tion cohorts was also quantified (Table 4), indicating that the nomogram model was an 168

effective classifier in different subgroups. The nomogram calibration plots revealed mod- 169

erate prediction accuracy in the training and validation cohorts (Figure 3). Figure 4 shows 170

the DCA for the training and validation cohorts to predict the possibility of DPN. A farther 171

line from the model curve to the black and gray meant a better nomogram’s clinical value. 172”

Figure 2. ROC curves of nomogram in training (A) and validation (B) groups.

And

lines “The AUC of the nomogram was 0.763 for the training cohort (Figure 2), and the in- 164

ternal validation by 200 bootstrap resamplings was 0.760. The AUC of the nomogram was

0.755 for the validation cohorts (Figure 2)”

Figure 3. Calibration plots of the nomogram in the training (A) and validation (B) groups.

Figure 4. The decision curve analysis of the nomogram predicts the possibility of DPN in training 178

(A) and validation (B) groups. A farther line from the model curve to the black and gray meant a 179

better nomogram’s clinical value.

Table 4. ROC analysis in different subgroups

*) This important part should be deeply improved. More information should be provided about the use of the nomograms to provide predictions. Please add some examples here or in appendix to illustrate their usage.

*) Are the AUC values 0.763 for the training cohort and 0.755 for the validation cohorts so different ? Please explain better

*) The figure captions should be more informative.

*) The quality of plots should be improved.

Lines: “3.3. Relationship Between Variables and Ncss Parameters 183

Correlation analysis showed that age, duration, BMI, SUA, HbA1c, and FT3 were 184

correlated with NCSs parameters in total patients (Supplementary table 1). In multiple 185

linear regression analysis (Supplementary table 2), sex, age, duration, BMI, SUA, HbA1c, 186

and FT3 were significantly associated with NCSs parameters. 18”

*) Please clarify in a better way and provide the claimed correlations within the main text.

Lines: “Our study still had some limitations. First, this was only a cross-sectional database 239
recruited from patients with T2DM in a single hospital, which can not represent commu- 240
nities or other hospitals. And the causality in this cross-sectional study is very limited. 241
Second, the present study did not collect information on other complications of T2DM and 242
genetic markers. However, these tests may increase the cost-effectiveness of DPN screen- 243
ing and other clinical indicators. Finally, a multicenter external validation is warranted to 244
evaluate the prediction performance of our nomogram. “

*) The authors should further discuss how the reported limitations could limit the statistical significance of their work. If “the causality in this cross-sectional study is very limited” are the claimed main results reliable ? If “ a multicenter external validation” should be still provided, what is the current value of this work ? Is it only a “proof of concept” ? Please explain better.

Author Response

Responses to the Reviewer 2 
Question 1: Title: “A Nomogram for Predicting the Possibility of Peripheral Neuropathy Among Type 2 Diabetes Mellitus Patients”. In this work the authors provide a nomogram to predict the probability of DPN tailored to individual patients. The authors claim that this predictive nomogram showed good discriminatory strength and moderate clinical value in training and validation cohorts. In particular, this nomogram was based on male, age, duration, BMI, SUA, HbA1c, and FT3 to provide a user-friendly and convenient tool for clinical practice. General comment: Although the topic of this work is interesting and the use of a nomogram may assist clinicians in a favourable way, this work should be reworked to improve its quality and impact. In particular, the readability of the main text should be enhanced providing, at the beginning of the main text, a table where all the used acronyms are described without a big effort of the readers, who is now forced to go back and forth to find the definition of each acronyms inside the main text.
Answer: Thank you very much for your careful review! As you suggested, we have provided Table1 with all the used acronyms in the paper.
Question 2: Then, the statistical methods used to build nomograms should be better explained for interested readers without of strong statistical background. 
Answer: Thank you very much for your careful review and insightful comments. This point is excellent! According to your suggestion, we have modified the statistical methods for interested readers without of strong statistical background in the materials and methods section.
Materials and Methods section (page 4, line 134): The validation process of the nomogram model involved two steps. First, the area under the curve (AUC) of the receiver-operating characteristic (ROC) was used to evaluate the nomogram's discrimination capacity in the training cohort. And we also conducted the AUC with a 95% CI using 500 bootstrap resamplings for internal validation. A calibration curve was assessed graphically by smoothing a scatter plot of the predicted and actual probabilities. Decision curve analysis (DCA) was performed to determine the clinical net benefit of the model. Second, the ROC, calibration curve, and DCA were also performed in the validation cohort. Correlations between variables in the nomogram and NCSs parameters were assessed using Pearson's or Spearman's test in total patients. Then multiple linear regression analysis was executed to identify the relationship between variables in the nomogram and NCSs parameters. Two-sided P < 0.05 was considered statistically significant.
Question 3: In addition, the use of nomograms should be also explained in a better way through examples of predictions. 
Answer: Thank you very much for your valuable advice. We appreciate it very much!
We have provided an example in the results section as follows:
Results section (page 6, line 177): For example, a 70-year-old (60 points) male (20 points) patient suffers from 5 years history of type 2 diabetes (10 points), has 30 kg/m2 of BMI (10 points), 500 umol/L of UA (30 points), 7 of HbA1c (10 points) and 8 pmol/L of FT3 (50 points) has a total score of 190 points. The estimated probability of DPN for this patient is slightly less than 50%.
Question 4: Finally, the value of this work is not totally clear since some important limitations are reported by the authors in the Discussion section.
Answer: Thank you very much for your suggestion. We have reworked the limitations in the discussion section as follows:
Discussion section (page 10, line 207): Our study still had some limitations. First, this was just a single-center study and was only suitable for patients with type 2 diabetes, which could not be generalized to communities or patients with type 1 diabetes. Second, the present study did not collect information on other complications of T2DM and genetic markers. However, these tests may increase the cost-effectiveness of DPN screening and other clinical indicators.
Question 5: Some detailed comments: Lines: “The study was approved by the ethics committee of the First Affiliated Hospital of Wenzhou Medical University (NO.KY2021-R141). Written informed consents were obtained from patients or their representatives “*) The informed consent should be obtained by all people who are involved in the experimental trials. It is not clear who “their representative” may be. Please explain.
Answer: Thanks very much for such a careful review. We have corrected the informed consent to “Written informed consents were obtained from patients and their relatives.”
Question 6: Lines: “2.4. Statistical Analysis Continuous variables described as the means with standard deviations or medians with interquartile ranges were compared between training and validation groups using Students’t-tests or the Kruskal–Wallis test. Categorical variables reported as percentages were compared between the two groups using Chi-square tests. First, univariate analysis and univariate logistic regression were performed to identify the risk factors of DPN in the training group. Then the clinical variables reported as essential factors in previous studies were also selected. Secondly, the stepwise multivariate logistic regression analyses were performed using variables with P < 0.05 from univariate analyses and critical clinical variables. A nomogram was established based on independent factors selected by multivariate logistic regression analyses of the training cohort. The area under the receiver operating characteristics curve (AUC) was used to evaluate the nomogram's discrimination capacity. And we also conducted the AUC with a 95% CI using 200 bootstrap resamplings for internal validation. A calibration curve was assessed graphically by smoothing a scatter plot of the predicted and actual probabilities. Decision curve analysis (DCA) was performed to determine the clinical net benefit of the model. The performance of the nomogram was further evaluated in the validation cohort via receiver-operating characteristic (ROC), calibration curve, and DCA. Correlations between independent factors and NCSs parameters were assessed using Pearson's or Spearman's test and multiple linear regression analysis. Two-sided P < 0.05 was considered statistically significant. All the statistical analyses were performed with statistical packages R (http://www.R-project.org) and SPSS Version 24.0 (SPSS Inc., Chicago, IL, USA).” These lines are not too clear and should be reworked in order to better clarify the statistical procedures used in this work. In particular, lines “Continuous variables described as the means with standard deviations or medians with interquartile ranges.” What are the variables assessed through their mean and standard deviation? What are the variables assessed through their medians and interquartile ranges. Why they are different? What is the test used to investigate the nature of these variables (paramateric, nonparametric) and of their distributions (normal, not normal).
Answer: We performed the Shapiro–Wilk test to check whether the data conform to the normal distribution. If the variable conforms to the normal distribution, we assessed it through their mean and standard deviation, using Students’t-tests, if not, we assessed it through medians and interquartile ranges, using Students’t-tests the Kruskal–Wallis test.
Question 7: Lines: “compared between training and validation groups using Students’t-tests or the Kruskal–Wallis test”. When the Student and the Kruskal-Wallis tests were used? What were the main hypotheses to do this? A clear distinction between parametric and not parametric tests should be provided together with the list of all the variables involved in these test (with the explicit indication of their nature).
Answer: Parameter test: To estimate or test the population parameters under the assumption of normal distribution. Non-parametric test: it does not depend on the specific form of the overall distribution and whether the test distribution is the same. We performed the Shapiro–Wilk test to check whether the data conform to the normal distribution. If the variable conforms to the normal distribution, we used the Student tests, if not, we used the Kruskal-Wallis tests.
Variables    Training cohort (n =519)    Validation cohort (n =259)    Statistical approach
Age (years)    57.76 ± 12.95    58.97 ± 12.49    Student tests
Duration (years)    10(4-12)    10(5-15)    Kruskal-Wallis tests
BMI (kg/m2)    24.21 ± 3.49    24.23 ± 3.60    Student tests
FPG (mmol/L)    7.95 ± 2.90    8.10 ± 3.69    Student tests
UA (umol/L)    329.04 ± 101.10    326.47 ± 98.71    Student tests
TC (mmol/L)    4.79 ± 1.35    4.81 ± 1.48    Student tests
TG (mmol/L)    1.91 ± 1.74    1.89 ± 1.47    Student tests
HDL-C (mmol/L)    1.02 ± 0.29    1.05 ± 0.43    Student tests
LDL-C (mmol/L)    2.61 ± 0.98    2.58 ± 1.03    Student tests
FIB (g/L)    3.79 ± 1.24    3.71 ± 1.05    Student tests
NLR    2.36 ± 1.41    2.45 ± 1.62    Student tests
PLT (×109/L)    221.10 ± 64.93    222.26 ± 61.99    Student tests
HbA1c (%)    9.32 ± 2.38    9.42 ± 2.82    Student tests
TSH (mIU/L)    1.34 (0.91-2.03)    1.31(0.86-1.92)    Kruskal-Wallis tests
FT4 (pmol/L)    11.24 ± 2.33    11.34 ± 2.11    Student tests
FT3 (pmol/L)    4.64 ± 0.79    4.67 ± 0.84    Student tests

Question 8: Lines: “Categorical variables reported as percentages were compared between the two groups using Chi-square tests. First, univariate analysis and univariate logistic regression were performed to identify the risk factors of DPN in the training group. Then the clinical variables reported as essential factors in previous studies were also selected. Secondly, the stepwise multivariate logistic regression analyses were performed using variables with P < 0.05 from univariate analyses and critical clinical variables.” Please explain in detail the comparison through the Chi-square test and the main hypotheses.
Answer: Thank you very much for your suggestion. If the sample size is greater than 40 and the theoretical frequency in each cell should not be less than 5, the Chi-square test can be used. When the sample size is greater than 40, but there is 1≤Theoretical frequency < 5, the chi-square value needs to be corrected. When the sample content is less than 40 or the theoretical frequency is less than 1, the Fisher exact method can only be used to calculate. We have explained more detail in the materials and methods section. 
Materials and Methods section (page 3, line 113): We performed the Shapiro–Wilk test to check whether the variable conforms to the normal distribution. Continuous variables with normal distributions were described as the means with standard deviations, and continuous variables with skewed distribution were described as the medians with interquartile ranges. Then the continuous variables were compared between two groups using the Students’t-tests or the Kruskal–Wallis test. Categorical variables reported as frequencies (percentages) were compared between the two groups using Chi-square tests.
Question 9: Please explain in detail the “univariate analysis” and the “univariate logistic regression” together with the main hypotheses.
Answer: Thank you very much for your suggestion. We have added the explanation in the materials and methods section. 
Materials and Methods section (page 4, line 121): All variables were respectively performed univariate analysis between the DPN group and the non-DPN group, and univariate logistic regression to identify the risk factors of DPN in the training cohort.(“Statistical Analysis” section)
Question 10: Please explain in a better way and provide all the needed details for the lines: “Secondly, the stepwise multivariate logistic regression analyses were performed using variables with P < 0.05 from univariate analyses and critical clinical variables.”
Answer: Thank you very much for the suggestion. We have explained it better and provided all the details in the materials and methods section.
Materials and Methods section (page 4, line 126): Secondly, the stepwise multivariate logistic regression analyses were performed using variables with P < 0.05 from univariate logistic regression. According to previous literature, the clinical variables recognized as critical factors were also entered into multivariate logistic regression analyses. (“Statistical Analysis” section)
Question 11: lines: “A nomogram was established based on independent factors selected by multivariate logistic regression analyses of the training cohort. The area under the receiver operating characteristics curve (AUC) was used to evaluate the nomogram's discrimination capacity. And we also conducted the AUC with a 95% CI using 200 bootstrap resamplings for internal validation. A calibration curve was assessed graphically by smoothing a scatter plot of the predicted and actual probabilities.” Please explain in all details how the nomogram has been built. The authors should also explain why “we also conducted the AUC with a 95% CI using 200 bootstrap resamplings for internal validation”. Why just 200 bootstrap?
Answer: Variables with P values <0.05 in multivariable analysis were remained to establish a nomogram in the training group. A score for each variable was calculated based on the regression coefficient values. We have explained more detail in the “Statistical Analysis” part. To prevent internal validation of overfitting in the training cohort. Boostrap is a method for internal validation, mainly for evaluating predictive models. We modify and carry out 500 bootstraps. 
Question 12: Lines: “A calibration curve was assessed graphically by smoothing a scatter plot of the predicted and actual probabilities. Decision curve analysis (DCA) was performed to determine the clinical net benefit of the model. The performance of the nomogram was further evaluated in the validation cohort via receiver-operating characteristic (ROC), calibration curve, and DCA.” Please explain better in all details
Answer: We have explained more detail in the materials and methods section.
Materials and Methods section (page 4, line 138): A calibration curve was assessed graphically by smoothing a scatter plot of the predicted and actual probabilities. Decision curve analysis (DCA) was performed to determine the clinical net benefit of the model. Second, the ROC, calibration curve, and DCA were also performed in the validation cohort. We have explained more detail in the “Statistical Analysis” part.
Question 13: lines: “A total of 778 eligible patients with type 2 diabetes were randomly divided into a training cohort (n = 519) and a validation cohort (n = 259). About 65.7% and 72.2% of patients were diagnosed as DPN in the training and validation cohorts. The differences in baseline characteristics between the training and validation cohorts were insignificant (Table 1)”. Table 1 is not clear. Please improve together with the caption, which should be more descriptive.
Answer: Thank you very much for your careful review. We have changed table 1 to table 2 and changed the caption to “The comparison of baseline characteristics of the patients in training and validation cohorts.”
Question 14: Lines: “3.1. Univariate and Multivariate Analyses To identify the risk factors of DPN, a univariate analysis was performed between the DPN group and non-DPN groups in the training cohorts (Table 2). Significant differences in the following variables were obtained: male, age, duration, smoking, hypertension, SUA, FIB, HbA1c, and FT3”. Table 2 is not clear. Please improve together with the caption, which should be more descriptive.
Answer: Thank you very much for your careful review. We have changed table 2 to table 3 and changed the caption to “The comparison of baseline characteristics according to the presence of DPN and the univariate logistic regression analysis for DPN in the training cohort.”
Question 15: Table 3. Multivariate logistic regression analysis for DPN risk factors in the training cohort. See the previous comment. 
Answer: Thank you very much for your careful review. We have changed table 3 to table 4 and changed the caption to “Multivariate logistic regression analysis for risk factors associated with DPN in the training cohort.”
Question 16: Lines: “3.2. Nomogram Development and Validation Based on the multivariable logistic regression analysis, the nomogram was developed for predicting the possibility of DPN by using male, age, duration, BMI, SUA, HbA1c, and FT3 in the training cohort (Figure 1). Each variable's position on the corresponding “Points” axis was identified to predict the DPN's probability. Then the users added up the score of each variable to acquire the total score. By drawing a vertical line from the total points axis to the bottom scale to assess the probability of DPN.” And lines: “Figure 1. The nomogram prediction model for DPN risk. The position of each variable on the corresponding “Points” axis was identified. Then the users added up the score of each variable to acquire the total score. By drawing a vertical line from the total points axis to the bottom scale to assess the probability of DPN. DPN, Diabetic peripheral neuropathy; BMI, body mass index; SUA, serum uric acid; HbA1c, glycated hemoglobin; FT3, free triiodothyronine.” This important part of the main text should be reworked to improve its clarity. In particular, it is not clear how the single scales correlated to the total score. The authors should provide an example to allow the interested readers to better understand the procedure. The authors may use arrows as standard in nomograms.
Answer: Thank you very much for your valuable advice. We appreciate it very much!
We have modified this part and provided an example in the results section as follows:
Results section (page 6, line 160): Based on the multivariable logistic regression analysis, the nomogram was developed for predicting the possibility of DPN by using male, age, duration, BMI, UA, HbA1c, and FT3 in the training cohort (Figure 1). Data were collected for patients with type 2 diabetes, and the position of each variable on the corresponding axis was confirmed. A vertical line was drawn from the top “Points” axis to each variable’s position for collecting the score of variable value.’’Then the users added up the score of each variable to acquire the total score on “Total Points”. By drawing a vertical line from the total points axis to the bottom scale to assess the probability of DPN. For example, a 70-year-old (60 points) male (20 points) patient sufferers from 5 years history of type 2 diabetes (10 points), has 30 kg/m2 of BMI (10 points), 500 umol/L of UA (30 points), 7 of HbA1c(10 points) and 8 pmol/L of FT3 (50 points) has a total score of 190 points. The estimated probability of DPN for this patient is slightly less than 50%.
Question 17: Lines: “The AUC of the nomogram was 0.763 for the training cohort (Figure 2), and the internal validation by 200 bootstrap resamplings was 0.760. The AUC of the nomogram was 0.755 for the validation cohorts (Figure 2), presenting with good prediction. Furthermore, the predictive performance of the nomogram model for different subgroups in the validation cohorts was also quantified (Table 4), indicating that the nomogram model was an effective classifier in different subgroups. The nomogram calibration plots revealed moderate prediction accuracy in the training and validation cohorts (Figure 3). Figure 4 shows the DCA for the training and validation cohorts to predict the possibility of DPN. A farther line from the model curve to the black and gray meant a better nomogram’s clinical value.” Figure 2. ROC curves of nomogram in training (A) and validation (B) groups. And lines “The AUC of the nomogram was 0.763 for the training cohort (Figure 2), and the internal validation by 200 bootstrap resamplings was 0.760. The AUC of the nomogram was 0.755 for the validation cohorts (Figure 2)” Figure 3. Calibration plots of the nomogram in the training (A) and validation (B) groups. Figure 4. The decision curve analysis of the nomogram predicts the possibility of DPN in training (A) and validation (B) groups. A farther line from the model curve to the black and gray meant a better nomogram’s clinical value. Table 4. ROC analysis in different subgroups. This important part should be deeply improved. More information should be provided about the use of the nomograms to provide predictions. Please add some examples here or in appendix to illustrate their usage.
Answer: Thank you very much for your valuable advice. We appreciate it very much!
We have provided an example in the results section as follows:
Results section (page 6, line 177): For example, a 70-year-old (60 points) male (20 points) patient suffers from 5 years history of type 2 diabetes (10 points), has 30 kg/m2 of BMI (10 points), 500 umol/L of UA (30 points), 7 of HbA1c (10 points) and 8 pmol/L of FT3 (50 points) has a total score of 190 points. The estimated probability of DPN for this patient is slightly less than 50%.
Question 18: Are the AUC values 0.763 for the training cohort and 0.755 for the validation cohorts so different? Please explain better 
Answer: The nomogram was established based on the training cohort. To prevent overfitting of the model, we further validated in the validation cohort. There are still some different baseline characteristics between the training and validation cohorts, such as duration (P = 0.061), which may make the ROC different.
Question 19: The figure captions should be more informative.
Answer: Thank you very much for your careful review and insightful comments! We have improved the captions to make them more informative.
Question 20: Lines: “3.3. Relationship Between Variables and Ncss Parameters Correlation analysis showed that age, duration, BMI, SUA, HbA1c, and FT3 were correlated with NCSs parameters in total patients (Supplementary table 1). In multiple linear regression analysis (Supplementary table 2), sex, age, duration, BMI, SUA, HbA1c, and FT3 were significantly associated with NCSs parameters.” Please clarify in a better way and provide the claimed correlations within the main text.
Answer: Thank you very much. We have clarified it in a better way in the results section.
Materials and Methods section (page 9, line 236): In supplementary table 1 and supplementary table 2, except for F-wave, the improvement of other NCSs parameters represented better nerve conduction function. In correlation analysis and multivariate linear regression analysis, age, duration, UA, and HbA1c were inversely correlated with NCSs parameters (except F-wave). Otherwise, BMI and FT3 positively correlated with NCSs parameters (except F-wave). 
Question 21: Lines: “Our study still had some limitations. First, this was only a cross-sectional database recruited from patients with T2DM in a single hospital, which can not represent communities or other hospitals. And the causality in this cross-sectional study is very limited. Second, the present study did not collect information on other complications of T2DM and genetic markers. However, these tests may increase the cost-effectiveness of DPN screening and other clinical indicators. Finally, a multicenter external validation is warranted to evaluate the prediction performance of our nomogram.” The authors should further discuss how the reported limitations could limit the statistical significance of their work. If “the causality in this cross-sectional study is very limited” are the claimed main results reliable? If “a multicenter external validation” should be still provided, what is the current value of this work? Is it only a “proof of concept”? Please explain better.
Answer: Thank you very much for the comments. We have reworked the limitations in the discussion section. 
Discussion section (page 10, line 307): Our study still had some limitations. First, this was just a single-center study and was only suitable for patients with type 2 diabetes, which could not be generalized to communities or patients with type 1 diabetes. Second, the present study did not collect information on other complications of T2DM and genetic markers. However, these tests may increase the cost-effectiveness of DPN screening and other clinical indicators.

Reviewer 3 Report

The authors diagnosed diabetic polyneuropathy based on nerve conduction studies. They referenced that diagnosis to create a nomogram that predicts DPN based on clinical background. The reviewer thinks that the use of NCS is much appreciated. The nomogram may be useful for inexperienced healthcare providers in rural areas with limited healthcare resources. However, as DPN is present in about two-thirds of diabetic patients, this nomogram is of limited benefit. This paper indicates that the presence of DPN should be considered in older, hyperglycemic patients with a long duration of diabetes. It is common knowledge for diabetologists and does not provide new knowledge.

[Major]

Please include the presence of dyslipidemia in each group.

Please check multicollinearity in regression analysis. In particular, the authors should consider BMI and T3, which would correlate with age.

[Minor]

As the authors do not present a urinalysis, a “serum” of SUA is unnecessary.

As the current cohort includes only 18 years and older, please correct the age line in the nomogram.

Please correct “golden standard” to “gold standard.”

Author Response

Responses to the Reviewer 3
Question 1: The authors diagnosed diabetic polyneuropathy based on nerve conduction studies. They referenced that diagnosis to create a nomogram that predicts DPN based on clinical background. The reviewer thinks that the use of NCS is much appreciated. The nomogram may be useful for inexperienced healthcare providers in rural areas with limited healthcare resources. However, as DPN is present in about two-thirds of diabetic patients, this nomogram is of limited benefit. This paper indicates that the presence of DPN should be considered in older, hyperglycemic patients with a long duration of diabetes. It is common knowledge for diabetologists and does not provide new knowledge. [Major] Please include the presence of dyslipidemia in each group.
Answer: Thank you very much for your careful review and insightful comments. This point is excellent! According to your suggestion, we have added “dyslipidemia” to the tables in the results section.
Question 2: Please check multicollinearity in regression analysis. In particular, the authors should consider BMI and T3, which would correlate with age.
Answer: Thank you for your valuable comment! We has used stepwise regression to reduce collinearity. Meanwhile, we calculated the VIF of each variable in the Logistics regression model, which was all < 5, as shown in the figure below, indicating no severe collinearity.

Question 3: [Minor] As the authors do not present a urinalysis, a “serum” of SUA is unnecessary.
Answer: Thanks very much for such a careful review. We have revised “SUA” into “UA”.
Question 4: As the current cohort includes only 18 years and older, please correct the age line in the nomogram.
Answer: We are very sorry that we failed to correct the age line in the nomogram for more than 18 years due to the limitation of the drawing tool.
Question 5: Please correct “golden standard” to “gold standard.”
Answer: Thank you for your suggestion. We have corrected it.

Round 2

Reviewer 1 Report

Dear Authors,

I read your point-by-point answer. My comment was satisfactorily met. I hope that scientific soundness of your articles and interest for readers  have improved.

Author Response

Thanks very much for your valuable comments and suggestions.

Reviewer 2 Report

Although the text provided by the authors is not totally clear (the revision should be provided in color and the corrected lines deleted)  It seems that the authors responded to the comments of this reviewer. However, the global readability of the work could be still improved. Figure 1 should be still improved and the calculation of the total score should be better presented through examples. Figure 3 could be still improved by explaining the meaning of the labels (and unit of measure) on all the main axes. Figure 4 is still not clear. Please improve the caption.

Author Response

Question 1: Although the text provided by the authors is not totally clear (the revision should be provided in color and the corrected lines deleted) It seems that the authors responded to the comments of this reviewer. However, the global readability of the work could be still improved.

Answer: Thanks very much for your suggestions. We have tried our best to improve the work (see the paper).

Question 2: Figure 1 should be still improved and the calculation of the total score should be better presented through examples.

Answer: Thanks very much for your suggestions. We have provided an example in the caption of Figure 1.

Question 3: Figure 3 could be still improved by explaining the meaning of the labels (and unit of measure) on all the main axes.

Answer: Thanks very much for your suggestions. We have improved the quality of Figure 3 and added an explanation of the meaning of the labels on all the main axes in the caption.

Question 4: Figure 4 is still not clear. Please improve the caption.

Answer: Thanks very much for your suggestions. We have improved the quality of Figure 4 and the caption.

Reviewer 3 Report

The authors addressed all issues. 

Author Response

Thank you again.